# Authentication of Shenqi Fuzheng Injection via UPLC-Coupled Ion Mobility—Mass Spectrometry and Chemometrics with Kendrick Mass Defect Filter Data Mining

**DOI:** 10.3390/molecules27154734

**Published:** 2022-07-24

**Authors:** Xingdong Wu, Yaowen Liu, Zijia Zhang, Zhihuang Ou, Guoxiang Wang, Tengqian Zhang, Huali Long, Min Lei, Liangfeng Liu, Wenhua Huang, Jinjun Hou, Wanying Wu, De-an Guo

**Affiliations:** 1National Engineering Research Center of TCM Standardization Technology, Shanghai Institute of Materia Medica, Chinese Academy of Sciences, Shanghai 201203, China; wuxingdong@simm.ac.cn (X.W.); liuyaowen@simm.ac.cn (Y.L.); zijiazhang@simm.ac.cn (Z.Z.); s20-zhangtengqian@simm.ac.cn (T.Z.); longhuali@simm.ac.cn (H.L.); mlei@simm.ac.cn (M.L.); daguo@simm.ac.cn (D.-a.G.); 2School of Chinese Materia Medica, Nanjing University of Chinese Medicine, Nanjing 210029, China; 3Limin Pharmaceutical Factory, Livzon Group Limited, Shaoguan 512028, China; ouzhihuang@livzon.cn (Z.O.); wangguoxiang@livzon.cn (G.W.); liuliangfeng@livzon.cn (L.L.); huangwenhua01@livzon.cn (W.H.); 4Guangdong Corporate Key Laboratory of High-End Liquid Medicine R&D, Industrilization, Shaoguan 512028, China; 5University of Chinese Academy of Sciences, Beijing 100049, China

**Keywords:** chromatographic fingerprint, high performance liquid chromatography-tandem mass spectrometry, *Radix codonopsis*, *Radix astragali*, Shenqi Fuzheng injection

## Abstract

Nearly 5% of the Shenqi Fuzheng Injection’s dry weight comes from the secondary metabolites of *Radix codonopsis* and *Radix astragali*. However, the chemical composition of these metabolites is still vague, which hinders the authentication of Shenqi Fuzheng Injection (SFI). Ultra-high performance liquid chromatography with a charged aerosol detector was used to achieve the profiling of these secondary metabolites in SFI in a single chromatogram. The chemical information in the chromatographic profile was characterized by ion mobility and high-resolution mass spectrometry. Polygonal mass defect filtering (PMDF) combined with Kendrick mass defect filtering (KMDF) was performed to screen potential secondary metabolites. A total of 223 secondary metabolites were characterized from the SFI fingerprints, including 58 flavonoids, 71 saponins, 50 alkaloids, 30 polyene and polycynes, and 14 other compounds. Among them, 106 components, mainly flavonoids and saponins, are contributed by *Radix astragali*, while 54 components, mainly alkaloids and polyene and polycynes, are contributed by *Radix codonopsis*, with 33 components coming from both herbs. There were 64 components characterized using the KMDF method, which increased the number of characterized components in SFI by 28.70%. This study provides a solid foundation for the authentification of SFIs and the analysis of its chemical composition.

## 1. Introduction

The unambiguous identification of chemical composition is important for the authentification of natural product drugs, especially herbal injections. The Shenqi Fuzheng injection (SFI) is used to improve the quality of life of patients with lung and stomach cancer, especially with respect to symptoms like fatigue, laziness, and spontaneous perspiration. SFI was approved by the national medical products administration of the People’s Republic of China in 1999 (Drug Approval Number: Z19990065) [1]. SFI is comprised of an equal ratio of two herbal water extracts, *Radix codonopsis* (Dangshen, Co) and *Radix astragali* (Huangqi, As) [2]. It has wide clinical application in China and has been on China’s National Reimbursement Drug List (NRDL) since 2004.

Until now, the chimeric composition of SFI has been investigated using several methods. Eighty-one major constituents have been tentatively characterized by ultra-fast liquid chromatography (UFLC) coupled with electrospray ionization quadrupole time of flight mass spectrometry (ESI-QTOF). These constituents include organic acids, amino acids, oligosaccharides, alkaloids, nucleosides, phenylpropanoids, polyacetylenes, flavonoids, isoflavonoids, and saponins [2]. Of these constituents, three saccharides, fructose, glucose, and sucrose, were demonstrated to account for nearly 90% of the SFI’s dry weight (except for the NaCl added), as assayed using the HPLC-ESLD method [3]. Nine of the fifteen analytes, including major polyacetylenes, flavonoids, isoflavonoids, and saponins, were determined [3,4], but only 0.18–0.21% of the SFI could be quantitatively analyzed. There was still nearly 5% of the key chimerical composition of the SFI that could be characterized, mainly attributed to the secondary metabolites of *Radix codonopsis* (Dangshen, Co) and *Radix astragali* (Huangqi, As), which seriously hinders the authentification of the SFI.

The chromatographic profile (also called the chromatographic fingerprint) technique has been widely used in the quality control and authentication of herbal medicines [5] and food [6]. For SFI, two chromatographic fingerprints have been used for quality control purposes. One is used to control nucleosides, phenylpropanoids, polyacetylenes, and flavonoids at 245 nm, and the other is used to assay saponins using an evaporative light-scattering detector (ELSD). However, it was difficult and laborious to characterize SFI using two independent methods. The charged aerosol detector (CAD), as a universal and high sensitivity detector, has advantages for fingerprint analysis compared with an ultraviolet detector (UV) and ELSD [7]. The CAD can simultaneously detect analytes with or without ultraviolet absorption. Additionally, due to its universal response for most analytes, the chromatographic profile obtained with CAD can readily reflect the actual contents of all analytes in herbal medicines.

High-resolution mass spectrometry (HRMS) with ion mobility (IM) is a high dimensionality analysis tool that can provide abundant chemical composition information when coupled with liquid chromatography [8,9,10]. It has been successfully used for the deep chemical analysis of the root of *Panax notoginseng* [11], the stem and hooks of *Uncaria rhynchophylla* (Miq.) Miq.ex Havil [12], the tuber of *Gastrodia elata* Bl [9], and the bulbs of *Lilium brownii var. viridulum* Baker [13]. The collision cross section (CCS) value obtained by ion mobility provides separation with the potential to distinguish isomers for HRMS [9].

To obtain a high content of chemical structural information with HRMS, many data acquisition modes, such as data-dependent acquisition (DDA) and data-independent acquisition (DIA), including data post-processing methods, should be performed [14]. The fragmentation ions obtained by DDA and DIA modes are crucial for the elucidation of structures. This is especially true for DIA mode (MSE mode, all-ion fragmentation, or SWATH), which can provide lossless fragmentation information for all potential analytes in herbal medicines. With online or in-house library matching using software (such as UNIFI and Progenesis QI [15]), the structural information in DIA or DDA data can be quickly screened [16]. In addition, data post-processing methods such as mass defect filtering (MDF) [12] and Kendrick mass defect filtering (KMD) [17] can also be used to screen potential analytes with similar structures.

In this study, a single chromatographic fingerprint was developed to cover all secondary metabolites in the SFI and its intermediates by adopting the UHPLC-CAD approach. Then, combined with IM-QTOF or LTQ-Orbitrap-MS, all peaks in the chromatographic fingerprint were dissected in-depth with mass defect filtering (MDF) and Kendrick mass defect filtering (KMD). Additionally, the CCS values of all tentatively identified ions were also demonstrated for the first time. The resulting in-depth chemical information regarding the SFI may provide useful information for its authentication.

## 2. Results and Discussion

### 2.1. Development of Chromatographic Fingerprint of SFI by UHPLC-CAD

Nearly 5% of the dry weight in SFI is still unknown, which hinders its authentication. Most of these components come from the secondary metabolites of *Radix codonopsis* and *Radix astragali*. Thus, a solid-phase extraction method with non-polar divinyl benzenebased neutral polymeric sorbent was optimized and developed to enrich those secondary metabolites. The discarded part of SPE consistent of primary saccharides (fructose, glucose, and sucrose, representing approximately 88% of the dry weight in the SFI), amino acids (approximately 2% of the dry weight in the SFI), and nucleosides (unpublished data). The sample preparation method, including the SPE type, loading sample volume, eluent volume of water and methanol, and the flow rate of the eluent and solvent of the sample, were comprehensively optimized (unpublished data).

The enriched part of the SPE column was mainly composed of polyacetylenes, flavonoids, and saponins. It was challenging to profile these components sensitively and simultaneously in a single chromatogram. Due to the prominent advantage of the sensitivity of the CAD as a universal detector, it was adopted to construct the chromatographic fingerprint of secondary metabolites in the SFI (Figure 1A). The chromatographic system was optimized comprehensively, including different columns, mobile system, mobile gradient, flow rate, column temperature, and injection volume. The CAD parameters were also compared one by one. The method validation was evaluated based on precision obtained on different columns, instruments, and labs (data not shown).

To dissect the chemical component information from the chromatographic fingerprint of SFI, the developed UHPLC method was coupled to IM-QTOF-MS and LTQ-Orbitrap Velos to obtain the high-resolution *m*/*z* value, CCS value, and MS/MS fragmentation information. The typical base peak chromatograms (BPC) in the positive and negative modes are shown in Figure 1B,C. The corresponding chromatograms of Co_E and As_E are shown in the Appendix A.

### 2.2. Combining PMDF and KMDF for MS Data Mining of SFI

The chemical composition of natural products is complex and diverse, and comprehensively characterizing the chemical composition of natural products is still a considerable challenge. Due to the high sensitivity, high resolution, and high accuracy of high-resolution mass spectrometry (HRMS), the data collected with HRMS contain massive compound information, including a large amount of background interfering information. In addition, when using HRMS to collect chemical information on natural products, it is more challenging to analyze the structure of chemical components due to the in-source dissociation of compounds and the additive effects on compounds when solvents are added. Mass defect filtering (MDF) is a method of quickly and efficiently screening target components in complex systems based on the law between the precise mass range and the mass loss range of compounds of the same structural type. PMDF, derived from MDF, establishes a polygonal *m*/*z* window for potential target components by using the reported compound information to eliminate matrix interference. Using PMDF, target components could be screened more precisely in complex systems. KMDF is based on the theory raised by Edward Kendrick. He proposes that compounds, when sharing the same skeleton type but differing in one or more CH_2_ groups, will result in the same KMD value. This theory can be applied to quickly screen homologs in natural products and is currently primarily used to analyze lipids.

Previous literature indicates that the SFI contains many types of compound skeletons, including flavonoids, saponins, polyene and polyacetylene, and alkaloids. The literature also indicates that there are one or more CH_2_ group differences within each type of compound’s skeletons. Therefore, by combining these two data filtering methods, we can quickly characterize the same type of components and their homologs in the SFI. A total of 198 components were collected by retrieving the previous literature and sorted into flavonoids (50), saponins (79), alkaloids (30), and polyene and polyacetylene (39) based on their structures, and various PMDF windows of different components were created by the programmed formula using Excel (Appendix A).

The QI data were filtered according to the established PMDF window, and the results were screened for potential flavonoids (366), saponins (389), alkaloids (391), and alkenes/alkyne (396) components. The potential flavonoids, saponins, alkaloids, and alkene components were further filtered using the KMDF method, where the bias was set to ±5 ppm to characterize as many constituents as possible in the SFI. The results are shown in Figure 2A. Taking compounds 34 and 44 as examples, they have the same KMD value, indicating that compounds 34 and 44 might be the same type of compound. Moreover, the NKM values differed by 14, indicating one CH_2_ unit difference between compounds 34 and 44. Similarly, more types of components including flavonoids, saponins, alkaloids, and alkenes can be analyzed in the SFI using this strategy. The numbers of homologs screened using the KMDF method were 30, 16, 16, and 2, for flavonoids, saponins, alkaloids, and alkenes/alkynes, respectively. Unfortunately, the KMDF method could not precisely characterize the absolute configurations of isomers.

When comparing the CCS values acquired by UPLC-IMS-Q-TOF analysis with the CCS values retrieved from the literature [18], the CCS values of the same components characterized in the current study were consistent with previous literature within 1% variation. This indicated that the CCS values obtained in the current study are accurate and can be applied to future analysis. The KMD method and the CCS values were integrated to analyze the characterized components and the results are shown in Figure 2B. Figure 2B shows that the higher the molecular weight of the identified compound, the larger the CCS value. Additionally, the CCS values of the saponins were larger than those of other types of components. These data contribute to establishing a CCS database of characterized compounds which can provide a reference for identifying chemical components in the SFI.

### 2.3. Characterization of Compounds in the SFI

The SFI is composed of *Radix codonopsis* and *Radix astragali*. *Radix astragali* mainly contains flavonoids and saponins, while *Radix codonopsis* mainly contains alkenes/alkynes and alkaloids [19,20,21,22]. Studies have shown that the flavonoids and saponins in *Radix astragali* have anti-inflammatory, fatigue-alleviating, anti-tumor, and immunoregulatory activity [23,24], and the alkenes/alkynes and alkaloids in *Radix codonopsis* have antioxidant and anti-tumor activity [25,26]. To characterize more flavonoids, saponins, alkaloids, and alkene in SFIs, the components in the SFI were identified by comparing the components with in-house databases, the mass spectrometry data from the literature, and the reference substances’ chromatography, spectra, and retention time data. Then, diagnostic fragment ions were identified for different types of components by summarizing the fragment ion peaks of the identified components. Additionally, more homologs were explored using identified components with the KMDF method.

#### 2.3.1. Characterization of Flavonoids

Both *Radix codonopsis* and *Radix astragali* contain flavonoids. The predominant flavonoids in *Radix codonopsis* are flavonol, flavanones, and their glycosides [20], while *Radix astragali* has mainly isoflavones, flavonoids, isoflavones, pterocarpans, chalcones, and their glycosides [19]. By summarizing the pattern of flavonoids’ break bonds in MS, it was noted that flavonoids mainly underwent glycoside bond breakage, dehydration, CO_2_, CO, C-ring fracture, and retro-Diels–Alder (RDA) reactions. In this study, 58 flavonoids (compound **1**–**58**) were identified by in-house databases, MS data from the literature, and retention time and MS data obtained using reference substances. 

Compounds **1**, **2**, and **12** were presumed to be flavonols according to their MS/MS spectra. In compound **1**, the [M − H]^−^ ion at *m*/*z* 623.16 Da and the [M + HCOO]^−^ ion at *m*/*z* 669.17 Da were detected in the MS spectra. Between the two ions, *m*/*z* 623.16 Da produced *m*/*z* 461.11, *m*/*z* 299.06, and *m*/*z* 151.04 fragmented ions, which were derived from successive losses of two Glc and C-ring fractures, respectively, followed by dicarbonyl group. Compound **1** was identified as a complanatuside or its isomer after comparison with MS data from the literature [27]. In MS, compound **2** gave an [M − H]^−^ ion at *m*/*z* 609.15 Da in negative ion mode and generated a molecular formula of C_27_H_30_O_16_ with an error of −1.72 ppm. In MS/MS spectra, compound **2** showed three principal fragment ions at *m*/*z* 447.09, *m*/*z* 285.04, and *m*/*z* 193.05 Da, which were from successive losses of two Glc (162 Da) and a C-ring fracture (Appendix A), respectively. Through comparison of the mass spectrometric data from the literature, compound **2** was tentatively identified as kaempferol-3-O-sophoroside [28]. In the negative ion mode, the pseudo-molecular ion peak of compound **12** was *m*/*z* 447.09 Da. It has one less Glc (162 Da) than compound **2**. In MS/MS spectra, three ion fragments were detected at *m*/*z* 285.04, 255.03, and 193.05 Da, which came from the consecutive loss of one Glc (162 Da) and HCOH, and the C-ring fracture of the flavonoids. Based on the information of the compound parent ion and its fragment ions, compound **12** was identified as astragalin.

Comparison with the reference standards’ retention time and the MS data information, compounds **6** and **44** were identified as pratensein-7-*O*-*β*-d-glucopyranoside and calycosin, respectively, and both belonged in the isoflavone group. Calycosin is an isoflavone glycoside, and the precursor ion [M − H]^−^ was *m*/*z* 283.06 in the negative ion mode. The main fragment ions in MS/MS were *m*/*z* 268.05, 239.04, 211.04, 195.05, and 183.05 Da, and these ions were derived from calycosin losing CH_3_ (15.02 Da), CHO (29.00 Da), O (15.97 Da), and 2CO (55.99 Da). The MS fragmentation pattern and secondary fragmentation plot of calycosin are shown in Appendix A. In negative ion mode, the precursor ions [M-H]^−^ of pratensein-7-*O*-*β*-d-glucopyranoside (**6**) was *m*/*z* 461.11 Da, and thus the molecular formula was C_22_H_22_O_11_. The MS/MS fragments derived from *m*/*z* 461.11 Da were *m*/*z* 299.06, *m*/*z* 284.03, and *m*/*z* 266.02 Da, created by successive losses of Glc, CH_3_, and H_2_O from pratensein-7-*O*-*β*-d-glucopyranoside.

Compound **47** was identified as isomucronulatol 7-*O*-glucoside through comparison with the literature’s MS data information, which belongs to the isoflavane group [29]. In the negative ion mode, the pseudo-molecular ion [M-H]^-^ was *m*/*z* 463.16 Da, and the fragment ions were *m/z* 301.11, 286.09, 271.06, and 147.05 Da in the MS/MS spectra (Appendix A). These fragment ions were derived from isomucronulatol 7-*O*-glucoside by consecutive losses of Glc, CH_3_, and CO, as well as C-ring fracture in the basic skeleton of isoflavane.

Three of the 58 flavonoids were identified using reference substances, and the results showed that the analyzed structures contained isoflavones, flavonols, and isoflavone components. Unfortunately, the various classifications of flavonoid components and the presence of many isomers among different types make it impossible to characterize the flavonoid components one by one.

#### 2.3.2. Characterization of Saponins

The saponins in the SFI are mainly derived from *Radix astragali* and *Radix codonopsis* herbs and most of them are tetracyclic triterpenes and pentacyclic triterpenes [20,27]. Pharmacological studies have shown that saponins have immunomodulatory [30], anti-tumor [31], antioxidant [32], and neuroprotective activities [33]. Therefore, it is crucial to characterize the saponin components in the SFI.

In negative ion mode, the molecular ion peaks [M − H]^−^ and additive ion peaks [M + HCOO]^−^ of compounds **98**, **100**, and **102** were at *m*/*z* 783.45 Da and *m*/*z* 829.46 Da, respectively, and the three were also isomers. Compounds **98**, **100**, and **102** were identified as astragaloside IV, astragaloside III, and isoastragaloside IV through comparison of the retention time and MS information using reference substances, and the three belonged in the tetracyclic triterpenoid group. In the MS/MS spectra, *m*/*z* 621.40 Da and *m*/*z* 489.36 Da peaks were detected in compounds **98**, **100**, and **102** fragments, and the two were derived by successive loss of one Xyl and one Glc from the precursor ions, respectively. Thus, the fragment ions *m*/*z* 621.40 Da and *m*/*z* 489.36 Da can be applied for the identification of tetracyclic triterpenoids in the SFI when their glycoside belongs to cycloastragenol.

The MS/MS spectra and possible MS fragment patterns of astragaloside IV are shown in Appendix A. In negative ion mode, compounds **71**, **74**, and **92**, and their precursor ions ([M − H]^−^ *m*/*z* 945.51 Da and [M + HCOO]^−^ *m*/*z* 991.52 Da) were detected, and they contained one more Glc than astragaloside IV. Further comparisons were made with the retention time and MS data of the reference substances, and compounds **71**, **74**, and **92** were identified as astragaloside VII, astragaloside VI, and astragaloside V, respectively. Compared to compound **71**, the molecular ion peaks of compound **91** had two less H atoms and MS/MS fragments than compound **71**. In negative ion mode, it was shown that compound **91** is a tetracyclic triterpene saponin with a skeleton type containing two less H atoms than cycloastragenol.

Furthermore, compounds **105**, **114**, and **117** were identified as rajanoside, isoastragaloside II, and cyclocephaloside II through comparison with reference substances. The fragment ions of compound **79** were [M − H]^−^ *m*/*z* 785.47 Da and [M + HCOO]^−^ *m*/*z* 831.48 Da, and its molecular formula was tentatively identified as C_41_H_70_O_14_ (−0.52 ppm). In the MS/MS spectra, fragment ions at *m*/*z* 623.42, 491.37, 473.33, and 415.32 Da were detected, which can be derived by successive losses of Xyl, Glc, H_2_O, and CO groups from the *m*/*z* 785.47 Da ion. After review of the literature, compound **79** was identified as cyclocanthoside E [33]. The MS/MS spectra and possible MS fragmentation patterns of cyclocanthoside E are shown in Appendix A.

Compound **106** precursor ions [M − H]^−^ *m*/*z* 941.51 Da and [M + HCOO]^−^ *m*/*z* 987.51 Da were identified as soasaponin I through comparison of the retention time and MS data with the reference substance. Soyasaponin I is a pentacyclic triterpenoid saponin, and its fragment ions were detected in the MS/MS spectra as *m*/*z* 795.46, *m*/*z* 615.40, and *m*/*z* 457.37. These fragment ions were derived from the successive losses of Gal, Glc, and C_6_H_6_O_5_ from soyasaponin I. The MS/MS spectra and possible MS fragmentation patterns of soyasaponin I are shown in Appendix A. Additiionally, the *m*/*z* 457.37 Da fragment ion can be used as a diagnostic ion to characterize pentacyclic triterpenoids in the SFI. In negative ion mode, compound **119**’s pseudo-molecular ion [M − H]^−^ and MS/MS fragment had two H less than soyasaponin I, indicating that compound **119** may be a compound containing a double bond in the soyasaponin I skeleton. Based on the in-house database, PMDF combined with the KMDF approach was further adopted, and **71** saponins were characterized in the SFI. Eleven of the **71** saponins were confirmed by comparison with the reference substances.

#### 2.3.3. Characterization of Alkaloids

Many studies have demonstrated that alkaloids in natural products have good biological activity [34,35]. For example, most analgesic drugs are solely made of alkaloid components. Though a small number of alkaloids have been reported in *Radix codonopsis*, few studies have been performed focusing on alkaloid components in the SFI. Therefore, it is necessary to characterize the alkaloid components in the SFI systematically.

Previous studies have shown that alkaloids are prone to obtain a proton in the positive ion mode and generate a [M + H]^+^ molecular ion peak with a better response [36]. Therefore, the characterization of alkaloids in the SFI was performed in positive ion mode. The ion peaks [M + H]^+^ at *m*/*z* 416.19 Da and [M + Na]^+^ at *m*/*z* 438.17 Da of compounds **127**, **132**, and **134** were detected in positive ion mode. Through comparison with the MS data in the literature, it was indicated that these ion peaks are codonopiloside A and its isomers [2]. The main fragment ions produced by codonopiloside A in MS/MS spectra were at *m*/*z* 398.18, *m*/*z* 254.14, *m*/*z* 236.13, *m*/*z* 218.12, *m*/*z* 205.09, and *m*/*z* 161.06 Da, and these ions were derived from codonopiloside A after successively losing one molecule of Glc, H_2_O, NH_2_CH_3_, and C_2_H_4_O. The possible fragment patterns of codonopiloside A in MS spectra are shown in Figure 3B. The *m*/*z* 398.18, *m*/*z* 254.14, *m*/*z* 236.13, *m*/*z* 218.12, *m*/*z* 205.09, and *m*/*z* 161.06 Da fragment ions can be applied as components of the diagnostic ion in *Radix codonopsis*, whose hydrogen in the N atom of pyrolidine alkaloids was substituted by a single CH_3_ group.

Compounds **153** and **158** are a pair of isomers and *m*/*z* 350.20 Da [M + H]^+^ ions were observed in positive ion mode. Through comparison with MS data from the literature, compounds **153** and **158** can be identified as codonopyrrolidium A and its isomers. Fragment ions at *m*/*z* 268.15, *m*/*z* 250.14, *m*/*z* 220.13, *m*/*z* 205.09, and *m*/*z* 161.06 Da were observed in the MS/MS spectra, and these fragment ions were derived by successive losses of C_5_H_7_O, H_2_O, NH(CH_3_)_2_, and C_2_H_4_O from the pseudo-molecular ion, respectively. According to the codonopyrrolidium A MS/MS fragment ions, their possible MS fragmentation patterns are shown in Figure 3D. The fragment ions at *m*/*z* 268.15, *m*/*z* 250.14, *m*/*z* 220.13, *m*/*z* 205.09, and *m*/*z* 161.06 Da can be used as diagnostic ions in the *Radix codonopsis*, whose hydrogens in the N atom of pyrolidine alkaloids are substituted by two CH_3_ groups. The precursor ions [M + H]^+^ of compounds **133** and **141** were detected at *m*/*z* 430.21 Da. In the MS/MS spectra, it can be observed that there are more CH_2_ fragments than those of codonopiloside A, indicating that compounds **133** and **141** might come from the triple CH_3_ replacement of the hydrogen on the N atom of codonopiloside A. The KMDF analysis showed that compound **136** had one more CH_2_ group than compound **130** and its pseudo-molecular ion peak was at [M + H]^+^ *m*/*z* 430.21 Da. Fragment ions at *m*/*z* 412.19, *m*/*z* 268.15, *m*/*z* 250.14, *m*/*z* 232.13, and *m*/*z* 205.09 Da were detected in the MS/MS spectra, which indicated that compound **136** was derived from the replacement of the H atom with two CH_3_ groups on the N atom of codonopyrrolidium A. The possible MS fragment pattern of compound **136** is shown in Figure 3C and no information regarding compound **136** can be retrieved in the SciFindern database, indicating that it is potentially a new compound.

In this study, 50 alkaloid components were characterized in the SFI using the PMDF combined with KMDF strategy, Most of the alkaloid components are pyrrolidine alkaloids from *Radix codonopsis*. However, few studies have been reported on alkaloids in *Radix codonopsis* and *Radix astragali*. The structure of the characterized alkaloids cannot be determined.

#### 2.3.4. Characterization of Polyacetylenes, Polyenes, and Their Glycosides

Polynephthalenes, polyenes, and their glycoside components are widely distributed among *Campanulaceae* plants and are the characteristic components of *Radix codonopsis*. Multiple unsaturated bonds make this type of component more active and prone to addition and oxidation reactions. Therefore, a large number of prototypes of such components can be characterized by the MS method.

Compounds **181** and **182** are a pair of isomers, with [M − H]^−^ and [M + HCOO]^−^ ions at *m*/*z* 423.19 and *m*/*z* 469.19 Da, respectively. In negative ion mode, fragment ions at *m*/*z* 261.13, *m*/*z* 221.07, *m*/*z* 179.06, and *m*/*z* 161.05 were detected in the MS/MS spectra, and they were derived from *m*/*z* 423.19 Da by losing one Glu, hexenyl fracture, [Glu-H]^−^, and [Glu-H_2_O-H]^−^. Through comparison of the MS data and the polarity of the compounds with the literature, compounds **181** and **182** can be inferred to be (*E*)-2-Hexenyl-*β*-sophorodide and (*Z*)-3-Hexenyl-*β*-sophorodide, respectively [37].

Compounds **201**, **202**, **204**, and **205** have the same pseudo-molecular ions [M − H]^−^ and addition ions in primary mass spectrometry [M + HCOO]^−^ *m*/*z* 395.17 and 441.18 Da, indicating that they are isomers of each other, and are inferred to lobetyolin through comparison with the literature mass spectrometry data and that of its isomer [2]. Fragment ions at *m*/*z* 305.12, *m*/*z* 233.12, *m*/*z* 215.11, *m*/*z* 179.05, and *m*/*z* 143.07 Da were detected in the MS/MS spectra, and they were derived from lobetyolin’s lipid chain breakage and losses of Glu, H_2_O, and CH_2_O, respectively. The MS/MS spectra and their possible fragment patterns of lobetyolin are shown in Figure 4B, indicating that *m*/*z* 305.12, 233.12, 215.11, 179.05, and 143.07 Da can be used as diagnostic ions for this class of components. Compounds **183**, **192**, **194**–**196**, **198**, and **200** in the MS spectra exhibited one more C_6_H_10_O_5_ group (*m*/*z* 162.05 Da) than lobetyolin, and they were presumably identified as lobetyolinin and its isomers. Furthermore, compound **203** had two more hydrogen atoms than lobetyolin, while the RDB value was 6.5 (lobetyolin, RDB = 7.5). This indicated that compound **203** might be codonopiloenynenoside A. Using the diagnostic ion method, a total of 30 compounds, including polyacetylenes, polyenes and their glycosides, were characterized from the SFI.

#### 2.3.5. Characterization of Other Components

In addition to flavonoids, saponins, alkaloids, polyacetylenes, and polyenes, there are also a small amount of phenylpropanin, lignans, and unsaturated fatty acids in SFI [2]. Other classes of ingredients in the SFI were characterized by comparing the MS data of chemical constituents in *Radix codonopsis* and *Radix astragali* plants to previous literature. The molecular ion peak [M − H]^−^ of compound **217** was *m*/*z* 327.22 Da, detected in negative ion mode, which was inferred to belong to 9,12,13-Trihydroxy-10,15-octadecadienoic acid using MS data from the literature [38]. This compound belongs to the unsaturated fatty acid group. Fragment ions at *m*/*z* 309.16, *m*/*z* 291.18, *m*/*z* 229.12, *m*/*z* 211.11, and *m*/*z* 171.10 Da were detected in the MS/MS spectra and they are derived from the continuous loss of H_2_O molecules and the breakage of fat chains from 9,12,13-trihydroxy-10,15-octadecadienoic acid. Their MS/MS fragment patterns are shown in Figure 4C. Similarly, compounds **217**–**224** were identified as unsaturated fatty acids. Compound **215** was identified as syringaresinol 4’-*O*-glucopyranoside through comparison of the retention time and MS data with those of the reference substance. Fragment ions at *m*/*z* 417.16, *m*/*z* 402.13, *m*/*z* 387.10, and *m*/*z* 371.15 Da were detected in the MS/MS spectra, derived by the losses of CH_3_ and HCOOH groups from the precursor ions, respectively. The MS/MS spectra of lobetyolin and their possible fragment patterns are shown in Figure 4D.

### 2.4. Analysis of Identified Compounds in SFI

To further analyze the source of the constituents in the SFI, the identified 233 compounds (Appendix A) were uploaded into the EHBIO (http://www.ehbio.com/test/venn/#/ (accessed on 1 June 2022) data analysis platform for analysis [39]. As shown in Figure 5A, 163 components were identified from *Radix astragali*, 111 components from *Radix codonopsis*, and 194 compounds were identified in the SFI using IMS-Q-TOF. Both *Radix astragali* and *Radix codonopsis* had five exclusive compounds in them and one exclusive component was identified in the SFI. Figure 5B shows that the identified components include 58 flavonoids, 71 saponins, 50 alkaloids, 30 polyenes, and polynephthyls, and 14 other types of compounds (such as unsaturated fatty acids, phenylpropanins, and lignans). Figure 5C showed that alkaloids, polyenes, and polynephthenes detected in SFI are mainly contributed by *Radix codonopsis*, while flavonoids and saponins are mainly contributed by *Radix astragali*. Figure 5D showed that the flavonoids (33) and saponins in the identified components mainly originated from *Radix astragali* (3), the alkaloids are from *Radix astragali* (25) and *Radix codonopsis* (25), and polyene and polynephthalene (25) components are mainly derived from *Radix codonopsis* herbs. These results lead to a detailed understanding of the origin and types of chemical components in the SFI, providing a reference to select quality control markers for SFI.

## 3. Materials and Methods

### 3.1. Reagents and Samples

A total of 18 standards were used for fragmentation behavior studies and confirmation of identification, including pratensein 7-*O*-glucopyranoside (CAS 36191-03-4, C_22_H_22_O_11_, MW. 462.12 Da; **06**), calycosin 7-*O*-*β*-d-glucoside (CAS 20633-67-4, C_22_H_22_O_10_, MW. 446.12 Da; **14**), calycosin (CAS 20575-57-9, C_16_H_12_O_5_, MW. 284.07 Da; **44**), 9-*O*-Methylnissolin 3-*O*-glucoside (CAS 94367-42-7, C_23_H_26_O_10_, MW. 462.15 Da; **40**), ononin (CAS 486-62-4, C_22_H_22_O_9_, MW. 430.12 Da; **32**), isomucronulatol 7-*O*-glucoside (CAS 94367-43-8, C_23_H_28_O_10_, MW. 464.17 Da; **47**), syringin (CAS 118-34-3, C_17_H_24_O_9_, MW. 371.14 Da), lobetyolin (CAS 136085-37-5, C_20_H_28_O_8_, MW. 396.18 Da; **200**), lobetyolinin (CAS 142451-48-7, C_26_H_38_O_13_, MW. 558.23 Da; **193**), astragaloside IV (CAS 84687-43-4, C_41_H_68_O_14_, MW. 784.46 Da; **98**), astragaloside III (CAS 84687-42-3, C_41_H_68_O_14_, MW. 784.46 Da; **100**), astragaloside II (CAS 84676-89-1, C_43_H_70_O_15_, MW. 826.47 Da; **105**), isoastragaloside II (CAS 86764-11-6, C_43_H_70_O_15_, MW. 826.47 Da; **114**), astragaloside I (CAS 84680-75-1, C_45_H_72_O_16_, MW. 868.48 Da; **127**), astragaloside VI (CAS 84687-45-6, C_47_H_78_O_19_, MW. 946.51 Da; **74**), cyclocephaloside II (CAS 215776-78-6, C_43_H_70_O_15_, MW. 826.47 Da; **117**), isoastragaloside I (CAS 84676-88-0, C_45_H_72_O_16_, MW. 868.48 Da; **127**), soyasaponin I (CAS 51330-27-9, C_48_H_78_O_18_, MW. 942.52 Da; **106**), isoastragaloside IV (CAS 136033-55-1, C_41_H_68_O_14_, MW. 784.46 Da; **101**). All standards were obtained from Shanghai Standard Technology Co. Ltd. The purity of all compounds met the requirement (>98%).

HPLC-Grade acetonitrile, methanol (Merck, Darmstadt, Germany), formic acid (FA), and deionized water (18.2 MΩ at 25 °C), prepared by the Millipore Alpha-Q water purification system (Millipore, Bedford, MA, USA), were used in the mobile phase for chromatographic separation and the extraction solvent. Leucine-enkephalin was purchased from Sigma-Aldrich (St. Louis, MO, USA).

The SFI (NO:190135) and the corresponding samples during the manufacture, including the single herb extract Co_E (NO: 190135-Co) and As_E (NO: 190135-As) were provided by Livzon Pharmaceutical Group Inc (Zhuhai, China).

### 3.2. Sample Preparation

The SFI sample was enriched with SPE (Bond Elut Plexa, 6 mL/200 mg, Agilent). After washing with water, the methanol eluent was collected and evaporated to dry. The residue was dissolved in 5.0 mL water with a final concentration 0.6 g/mL Co or As.

The other samples, including Co_E, and As_E, were prepared with the same procedure, and the concentration was adjusted to a final concentration 0.6 g/mL Co or As.

### 3.3. Chromatogram Fingerprint Obtained with UHPLC-CAD

The SFI was separated on an Agilent Poroshell 120 EC C18 column (150 mm × 4.6 mm, 2.7 μm) on an Ultimate ™ 3000 Standard Dual System (Thermo Fisher Scientific, San Jose, CA, USA) equipped with dual-gradient pumps, an autosampler, and a charged aerosol detector (CAD, Corona Veo). A binary mobile phase consisting of acetonitrile (A) and 0.1% formic acid in water (B) at a flow rate of 0.8 mL/min was adopted with a gradient elution program: 0–1 min, 12% (A); 1–13 min, 12–18% (A); 13–34 min, 18–25% (A); 34–48 min, 25–35% (A); 48–59 min, 35–70% (A); 59–59.1 min, 70–95% (A); 59.1–64 min; 95–95% (A); 64–65 min, 95–12% (A); 65–75 min, 12–12% (A). The column temperature was set at 25 °C and the CAD detector was optimized at a sampling frequency of 10 kHz, filter 3.6 s, and a low evaporator temperature (35 °C).

### 3.4. CCS Value Obtained from UHPLC -IM-QTOF

The CCS value and DIA analysis (MS^E^) were performed on a traveling wave ion-mobility hybrid quadrupole time-of-flight mass spectrometer with an ESI ion source (SYNAPT G2-Si, Waters, Wilmslow, UK) in resolution mode (~30,000 FWHM at *m*/*z* 500). The MS data were acquired from 100 to 1500 Da in positive and negative ion modes. The ESI source parameters were set as follows: capillary voltages of 0.5 kV (ESI+) or 2.5 kV (ESI−), a cone voltage of 40 V, a source offset of 60 V, a source temperature of 120 °C, desolvation gas flow of 800 L/h, desolvation gas temperature of 500 °C, and a cone gas (N_2_) flow of 20 L/h. The traveling wave ion-mobility mass separation parameters were set as follows: helium cell 95 mL/min; triWave, IMS wave velocity 650 m/s, wave height 20 V, transfer wave velocity 176 m/s; IMS wave velocity ramp starting at 80 m/s and ending at 1000 m/s. The MS and MS^E^ data scanning time was set at 0.5 s. Leucine enkephalin (LE; Sigma-Aldrich, St. Louis, MO, USA; 200 ng/mL) and sodium formate solvent (0.5 mmol/L) were used as the calibration standards. The CCS accuracy was calibrated with poly-dl-alanine solution (10 mg/L). MassLynx V4.1 software (Waters, Milford, MA, USA) was employed for data acquisition and processing.

### 3.5. Fragmentation Obtained from UHPLC-LTQ-Orbitrap

The MS/MS fragmentation information was obtained on the LTQ-Orbitrap Velos Pro-hybrid mass spectrometer (Thermo Fisher Scientific, San Jose, CA, USA) equipped with a heated electrospray interface (HESI) source in positive and negative ion mode. The parameters were set as follows: ion spray voltage, 3.8 kV (ESI+) or 2.7 kV (ESI−); collision energy, 30 eV; capillary temperature, 350 °C; source heater temperature, 300 °C. Nitrogen was used as the sheath gas and the auxiliary gas, and the gas flows were at 15 and 8 arbitrary units, respectively. The MS scan range of *m*/*z* 300–2000 was at the resolution of 30,000 (FWHM at *m*/*z* 400) as Event I. MS spectrum of top one intense ion from the parent list (PIL) was collected as Event Ⅱ. The most intense ion of MS^2^ would trigger MS^3^ fragmentation and was recorded as Event Ⅲ. The following parameters were used for dynamic exclusion: repeat count, 3; repeat duration, 2 s; exclusion list size, 50; exclusion duration, 30 s. Both CID/MS^2^ and CID/MS^3^ in normalized collision energies (NCE) were 35%. The resolution for Event II and III was set as 7,500 (FWHM at *m*/*z* 400). Xcalibur 4.1 software (Thermo Fisher Scientific, San Jose, CA, USA) was used to assist the data acquisition and processing.

### 3.6. Data Processing

According to previous literature, the organic small molecule compounds in Shenqi Fuzheng Injection are mainly composed of flavonoids, saponins, polyacetylenes and polyenes, alkaloids, and other components. To systematically characterize the different types of components in SFI, we imported the original MS file (QTOF MSE data) into QI, and the parameters of data processing were set as follows: the peak width was at 0.2 min; the minimum MS response was at 2000; the positive adduct ions were set as [M + H]^+^, [M + Na]^+^, [M + K]^+^, [2M + H]^+^, and [2M + Na]^+^; the negative adduct ions were set as [M − H]^−^, [M + COOH]^−^, [2M − H]^−^, and [2M + COOH]^−^, respectively. After retention time alignment, noise signal removal, and peak extraction, an .msp file was generated to integrate *m/z*, retention time, peak intensity, and CCS values in .scv format with data matrix and MS/MS data. The reported compounds in *Radix codonopsis* and *Radix astragali* were retrieved from literature, and an in-house database containing 198 identified compounds was established and imported into the UNIFI database for the following compound identification in the SFI. The compounds were collected, classified, and compiled with multiple formulas using Excel. PMDF windows were established for multiple components in the SFI to screen the potential flavonoids, saponins, polyacetylenes and polyenes, alkaloids, and other components in the obtained MS spectra for the SFI. To identify more components in the SFI, components were first identified with the PMDF method, then the KMDF method was applied to identify and recognize compounds with one or more CH_2_ groups [40]. The KMD formula is as follows:Kendrick mass (KM) = *m*/*z* × (Nominal mass of CH_2_ (14))/(Exact mass of CH_2_ (14.01565))(1)
KMD = (KM − NKM) × 1000(2)

In the above equations, *m*/*z* is the measured mass-to-charge ratio, and NKM is the nearest integer of KM.

In order to obtain more abundant and accurate information on the MS fragments from the identified components, the information of the fragments was further improved using LTQ-Orbitrap (Appendix A).

## 4. Conclusions

In the current study, a fingerprint of the secondary metabolites in the SFI was established by UHPLC-CAD and the fingerprints in the SFI were systematically characterized by chemical composition in combination with the IMS-Q-TOF and LTQ-Orbitrap techniques. Based on the characterization of the components in the SFI using an in-house database and reference substances, PMDF combined with the KMDF data processing method was used to mine the chemical composition information of the SFI. A total of 223 secondary metabolites were characterized from the SFI fingerprints, including 58 flavonoids, 71 saponins, 50 alkaloids, 30 polyene and polycynes, and 14 other types of compounds. Among them, the flavonoids and saponins in SFI are mainly contributed by *Radix astragali* and the enotylene components are mainly contributed by *Radix codonopsis*. Specifically, 106 components come from *Radix astragali*, including 33 flavonoids, 53 saponins, and 19 alkaloids. There were 64 components characterized using the KMDF method, which increased the characterization of the components in the SFI by 28.70%. This study provides a solid foundation for authentification analysis of the chemical compositions the SFI.

## Figures and Tables

**Figure 1 molecules-27-04734-f001:**
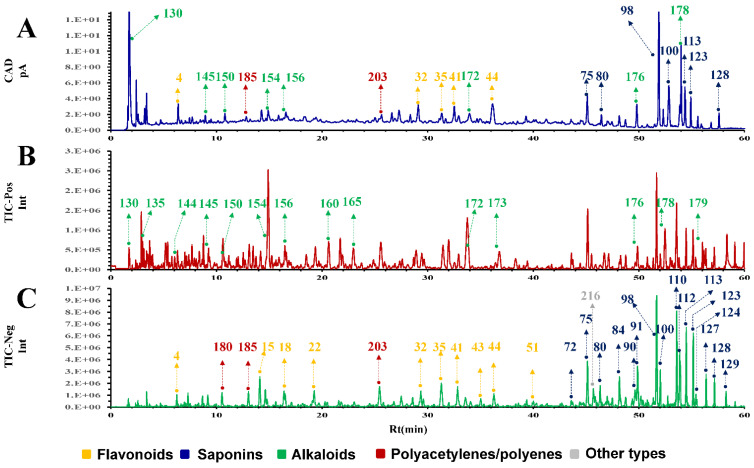
TIC diagrams in the positive and negative ion modes of UHPLC-CAD fingerprint established for the Shenqi Fuzheng injection (SFI). ((**A**), SFI fingerprint; (**B**), TIC diagram of SFI in positive ion mode; (**C**), TIC diagram of SFI in negative ion mode).

**Figure 2 molecules-27-04734-f002:**
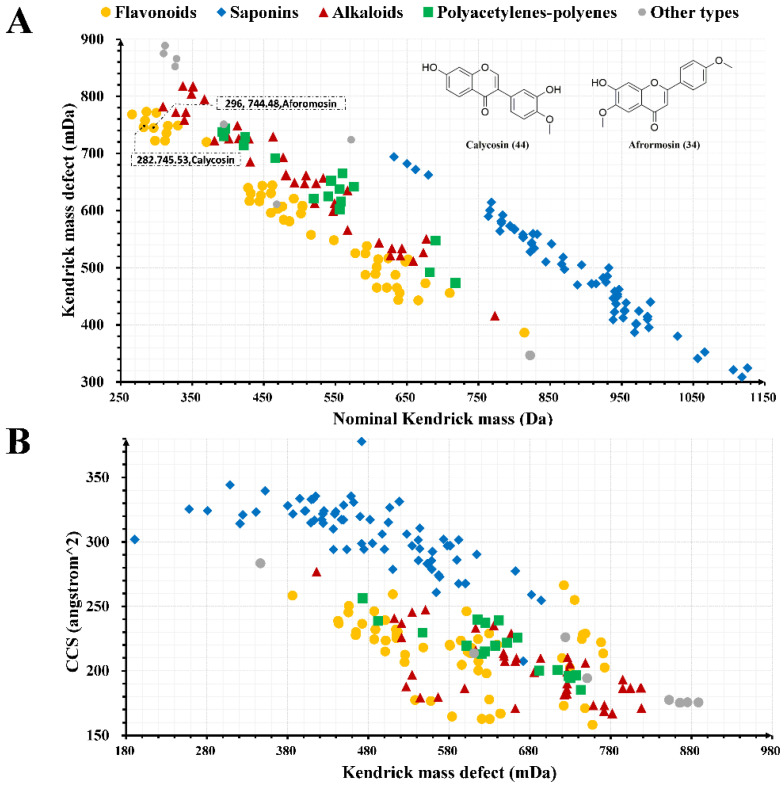
The KMDF scatter plot depicted the *m*/*z* values of different types of components in SFI and displayed the relationship between CCS values of various compound types and KMD. ((**A**), KMDF scatter plot of different types of chemical components in SFI; (**B**), diagram of different types of components and the CCS values).

**Figure 3 molecules-27-04734-f003:**
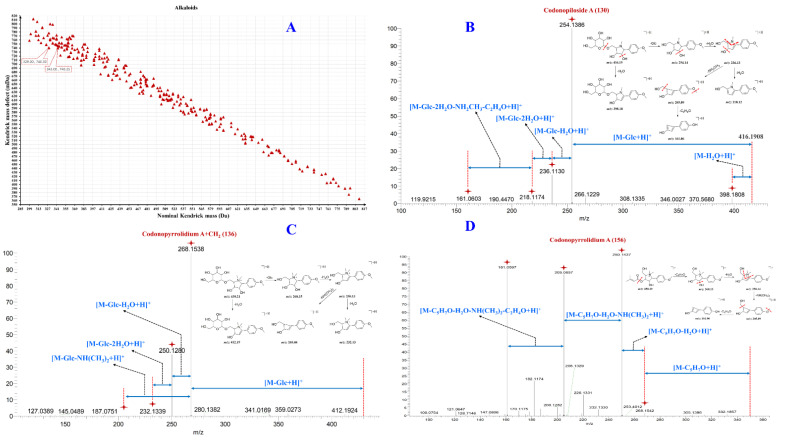
The KMDF scatter plot depicted the *m*/*z* values of potential alkaloids in SFI after PMDF treatment in Shenqi Fuzheng Injection (SFI) and the MS/MS spectra of the representative alkaloids and possible MS fracture patterns. ((**A**), KMDF scatter plot of potential alkaloids; (**B**), codonopiloside A; (**C**), codonopiloside A + CH_2_; (**D**), codonopyrrolidium A).

**Figure 4 molecules-27-04734-f004:**
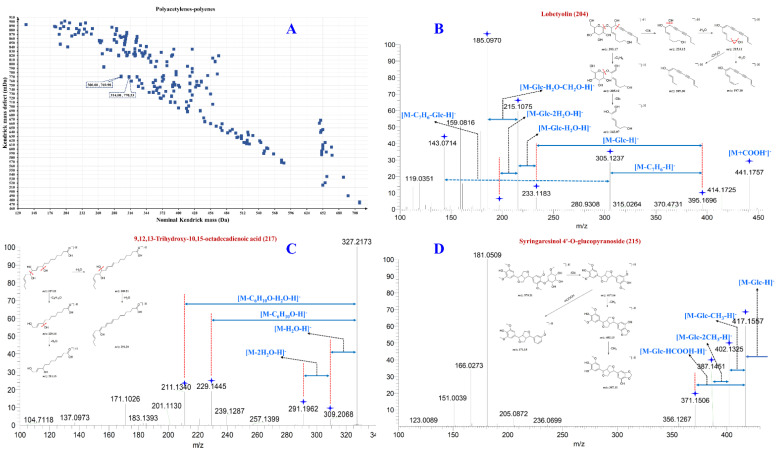
The KMDF scatter plot depicted the *m*/*z* values of potential polyenes and polynephthenes in Shenqi Fuzheng Injection (SFI), the MS/MS spectra of the representative polyene and polycynes, and the possible MS fracture patterns. ((**A**), KMDF scatter plot of potential polyene and polycynes; (**B**), lobetyolin; (**C**), 9,12,13-Trihydroxy-10-octadecadienoic acid; (**D**), syringaresinol 4′-*O*-glucopyranoside).

**Figure 5 molecules-27-04734-f005:**
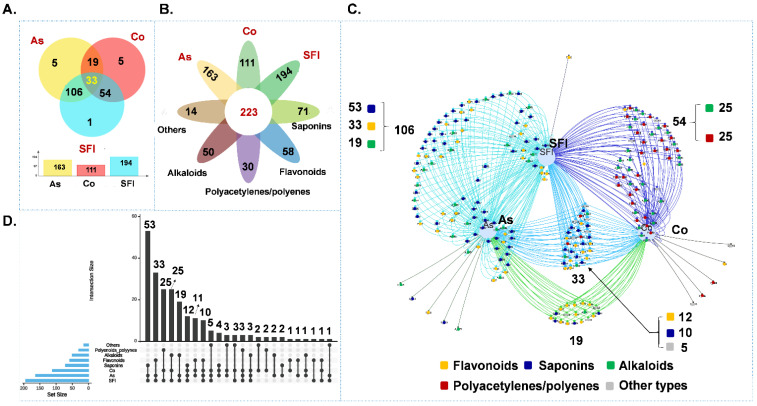
Various analyses show the source plants and composition of chemical components in Shenqi Fuzheng Injection (SFI). ((**A**), Venn diagram of the characterized components in SFI; (**B**), flower plot of the characterized chemical compositions in SFI; (**C**), Venn network of chemical constituents in SFI; (**D**), UpSet plot of chemical constituents in SFI).

## Data Availability

Not applicable.

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
