# Peer review of "Authentication of Shenqi Fuzheng Injection via UPLC-Coupled Ion Mobility—Mass Spectrometry and Chemometrics with Kendrick Mass Defect Filter Data Mining"

_molecules, 2022, doi:10.3390/molecules27154734_

Round 1

Reviewer 1 Report

In the manuscript the chemical composition of SFI was investigated by UHPLC-CAD. A combination of IM-QTOF or LTQ-Orbitrap-MS was used. The paper is well written, the linguage is fluent and concepts are clear. The topic is interesting and the scientific approch is rigorous and the manuscript is well organized.

The paper accomplishes the scope of Journal and specific section, therefore I recommend the publication after minor revision.

Sample preparation: were the xtraction condtion were chosen based on previous experiences? you add thr possible reference. The information regarding SFI sample is missing.

Reviewer 2 Report

The manuscript provides a comprehensive report of secondary metabolites in Shenqi Fuzheng Injection (SFI), an injection solution prepared from Radix Codonopsis (RC) and Radix Astragali (RA). IM-QTOF or LTQ-Orbitrap-MS coupled to UPLC were used to analyze the SFI solutions, and RC and RA root extracts. 223 secondary metabolites were identified in SFI, including 58 flavonoids, 71 saponins, 50 alkaloids, 30 polyene and polycynes, and 14 other types of compounds. The analyses revealed that among the 223 metabolites identified in SFI, 106 metabolites, mostly flavonoids and saponins, were unique to RA and 54 metabolites, mainly alkaloids and polyene were from RC. The other 33 metabolites were common to both RC and RA.  The fragments’ m/z values and CCS values of these metabolites were reported in the manuscirpt, which could be a valuable resource for natural product research and SFI authentication.

Overall, the manuscript was well organized and presented. However, English grammar errors and incorrect use of words can be found throughout the manuscript. For example, in the introduction, line 38, “Clear identification” is better replaced with “Unambiguous identification”. Line 55, “quantitative analyzed” should be “quantitatively analyzed”. Line 108 “university detector” should be “universal detector”. There are more similar errors in the manuscript. I suggest that the authors carefully review the manuscript to correct the errors or use professional English proofreading services.

Reviewer 3 Report

The manuscript entitled «Authentication of Shenqi Fuzheng Injection by UHPLC hyphenated ion mobility - mass spectrometry and chemometrics with Kendrick mass defect filter data mining» is devoted to the chemical composition of traditional medicine (TM) - SFI. The manuscript starts with quite informative introduction where main constituents of this medicine are qualitatively and quantitatively described. The analytical strategy used in this research employs several modern techniques to discover and identify flavonoids, saponins, alkaloids and other compounds in this TM. There is not much to criticize in this paper; however, some points need to be clarified to the readers.

Please discuss PMDF windows in more detail.

Please clarify (Part 3.6) what MS/MS data exactly was used LTQ-Orbitrap or QTOF MSE.

Please explain why such low values (voltages of 0.5 kV (ESI+) or 2.5 kV (ESI)) were applied.

Please carefully check the use of upper and lower indices for charges and numbers in CH3 CH2 H2O [M+H]+ etc.

Minor comments

Line 467 «(ESI-)» instead of «(ESI)»

Line 313 «….as codonopyrrolidium A and its isomer» instead of «isomers»

Line 173 «compounds» instead of «com-pounds»

Lines 488, 490-491 «MS2…. MS3» instead of «MS 2 …. MS 3»

Lines 500-501 «[M+H]+… [M+HCOO]- … » instead of «M+H…. M+COOH » etc.
